

# Blowing snow in East Antarctica: comparison of ground-based and space-borne retrievals

Alexandra Gossart[1], Stephen P. Palm[2,3], Niels Souverijns[1], Jan T.M. Lenaerts[4], Irina V. Gorodetskaya[5], Stef Lhermitte[6], and Nicole P.M. van Lipzig[1]

[1]Department of Earth and Environmental Sciences, KU Leuven, Leuven, Belgium
[2]Science Systems and Applications, Greenbelt, MD, USA
[3]NASA Goddard Space Flight Center, Greenbelt, MD, USA
[4]Department of Atmospheric and Oceanic Sciences, University of Colorado Boulder, Boulder CO, USA
[5]Centre for Environmental and Marine Studies, Department of Physics, University of Aveiro, Aveiro, Portugal
[6]Department of Geosciences and Remote Sensing, Delft University of Technology, Delft, the Netherlands

**Correspondence:** Alexandra Gossart (alexandra.gossart@kuleuven.be)

**Abstract.** Continuous measurements of blowing snow are scarce, both in time and space. Satellites now provide the opportunity to derive blowing snow occurrences, transport and sublimation rates over Antarctica. However, little ground truth is available to validate these retrievals. The recent application of ceilometers for detection of blowing snow frequencies provides an opportunity to validate the satellite retrievals of blowing snow frequencies at the Princess Elisabeth and Neumayer stations,

East Antarctica for the 2011-2016 time period. A routine to detect blowing snow occurrence from remote sensing ceilometers has been developed at those locations. Thanks to their ground-based location, ceilometers are able to detect blowing snow events in the presence of clouds and precipitation, which can be missed by the satellite, since optically thick clouds impede the penetration of the signal. This is important, since $\approx 90\%$ of blowing snow happens under cloudy conditions at Neumayer and Princess Elisabeth station and represent 30% of all cloudy conditions at both stations. Although both detection methods have

their limitations, 10% (4%) of the measurements at Princess Elisabeth (and Neumayer) are identified as blowing snow by the satellite but not by the ceilometer, likely due to differences in sensors, limitation of the surface identification by the satellite, or the spatial inhomogeneity of the blowing snow event. While the satellite blowing snow retrieval is a useful product, further investigation is needed to reduce the uncertainties on blowing snow frequencies associated with clouds.

## 1 Introduction

Blowing snow is a frequent phenomenon on the Antarctic Ice Sheet (AIS), occurring as often as 70% during winter (Palm et al., 2011). By redistributing snow, it is a locally and regionally important component of the surface mass balance (SMB). Wind-induced displacement of snow particles, dislodged from the surface and entrained in the near-surface air, can be a sink (referred to as erosion) or a source of mass (referred to as deposition).

Snow particles can creep on the snow surface, be in saltation between surface and atmospheric surface layer, or remain in

suspension in the atmospheric boundary layer (Leonard et al., 2012; Schlosser, 1999; Gallée et al., 2001). The displacement of particles is only a minor contributor to the integrated AIS SMB (Loewe, 1970; Dery and Yau, 2002; Lenaerts and van den



Broeke, 2012). However, the erosion of snow can have a strong impact on the SMB at a local or regional scale (Gallée et al., 2001; Dery and Yau, 2002; Lenaerts and van den Broeke, 2012; Groot Zwaaftink et al., 2013) through the displacement and relocation of snow particles, the erosion of snow cover and the exposure of blue-ice areas (Takahashi et al., 1988; Bintanja and van den Broeke, 1995). Since blue ice has a lower albedo than snow, more shortwave radiation is absorbed, enhancing surface

melt and water ponding at the surface. This mechanism is known as the wind-albedo feedback (Lenaerts et al., 2017).

Blowing snow is mainly driven by katabatic winds, which are gravity-induced atmospheric flows from the interior of the AIS towards the coast. In addition, particularly in coastal areas, blowing snow can be associated with synoptic-scale depressions that frequently reach the AIS coast (Gossart et al., 2017; Konig-Langlo and Loose, 2007). These weather systems originate from the Southern Ocean and bring precipitation onto the AIS (Souverijns et al., 2018), sometimes far into the interior (Schlosser,

1999; Hirasawa et al., 2013). During such events, the wind speed is relatively high and the snow particles from fresh snow accumulation are light, facilitating blowing snow. In addition to snow transport, the sublimation of blowing snow is an effective sink of AIS SMB (Kodama et al., 1985; Takahashi et al., 1992; Thiery et al., 2012; Dai and Huang, 2014): the particles suspended in the air offer a larger surface area to sublimation than those on the ground, resulting in more efficient sublimation (van den Broeke et al., 2004; Bintanja, 1998). Blowing snow sublimation was measured to represent 50 to 80% of the total

sublimation rate at two sites in East Antarctica (Bintanja and Reijmer, 2001).

Despite its important contribution to local AIS SMB, methods to measure blowing snow are still limited. Various techniques have been used ranging from mechanical traps or nets to acoustic sensors, optical sensors and macro photography techniques (eg,Trouvilliez et al. (2015); Amory et al. (2015); Leonard et al. (2012); Scarchilli et al. (2010)). These techniques are usually limited to a few sites and/or short time periods, and are scarce and uncertain due to the remoteness of the AIS and harsh

weather conditions. Satellite remote sensing has recently been used to retrieve blowing snow observations on the entire AIS. In particular, the Cloud-Aerosol LIdar with Orthogonal Polarization (CALIOP) on board the Cloud Aerosol Lidar and Infrared Pathfinder Satellite Observations (CALIPSO) has been used to design an algorithm that uses the CALIOP 20 Hz calibrated, attenuated backscatter profiles to derive blowing snow occurrence, layer height and optical depth (Palm et al., 2011), as well as to derive snow transport and snow sublimation rates over the full ice sheet since 2006 (Palm et al., 2017). However, satellite

blowing snow detection is hampered by the presence of (optically thick) clouds, which implies that the blowing snow retrieval is limited to clear-sky or optically thin cloud (< 2-3) conditions. Additionally, the vertical resolution of CALIPSO limits the detection to blowing snow layers to a minimum 30 meter thickness (Palm et al., 2017). And lastly, despite its potential for blowing snow detection, the CALIPSO product has not yet been validated.

The objective of this study is to compare the satellite retrievals of blowing snow from CALIPSO with novel ground-based

ceilometer observations of blowing snow. An algorithm has recently been developed to routinely detect the occurrence of blowing snow at two stations located in Dronning Maud Land, East Antarctica (Gossart et al., 2017). Ground-based remote sensing ceilometers offer the opportunity to derive blowing snow frequencies from the continuous measurement of attenuated backscatter. While it is very difficult to validate or even compare satellite-based retrievals of blowing snow with a point measurement on the ground, we believe the work presented here will help understand the limitations of the satellite measurements

and highlight the utility of both products.





In this paper, we will first describe the two data sets used in this study (Section 2). Section 3 reveals the extent of the agreement and disagreement of both methods and in Section 4 we conclude by describing the limitations of both detection methods for blowing snow, and provide some recommendations for future improvements.

## 2   Data and Methods

### 2.1   Satellite data

The satellite retrievals of blowing snow in this study are based on measurements from the CALIOP (Cloud-Aerosol LIdar with Orthogonal Polarization), which measures vertical profiles of clouds and aerosols at 1064 nm (backscatter intensity) and 532 nm (orthogonally polarized components). The 20 HZ CALIOP profiles equate to a horizontal resolution of 333 m along the satellite track. This lidar on board CALIPSO, launched in June 2006, is part of the A-train constellation of satellites and orbits the earth since 2006 at 705 km height (Winker et al., 2007, 2009). To retrieve blowing snow information, an algorithm was developed by Palm et al. (2011) and further refined in Palm et al. (2017). This algorithm uses the CALIPSO observations to derive blowing snow frequencies, layer heights, optical thicknesses, and also estimates the mass transport and the sublimation resulting from blowing snow, for the period 2006-2016 over the whole AIS. Here we limit our analysis to the period 2011 to 2016, for which we have concurrent ground-based observations. The blowing snow data are now available at the NASA Langley Atmospheric Science Data Center and includes satellite tracks (temporal and spatial coordinates) along with the derived presence of diamond dust, blowing snow occurrence (including a confidence level) as well as the wind speed and direction from the Goddard Earth Observing System Model, Version 5 (GEOS-5 (Gelaro et al., 2017)).

The original retrieval algorithm includes four steps (for an extensive description, see Palm et al. (2011, 2017)). First, the surface is detected. This implies that no (optically thick) clouds can be present, which would completely attenuate the backscatter signal. The algorithm identifies the surface level by starting 200m below the surface given by the digital elevation model (1 by 1km resolution GMTED Digital Elevation Model (DEM) used in the Goddard Earth Observing System 5 (GEOS-5)), and searches upward until a sufficiently strong return of the attenuated backscatter signal indicates the surface. Second, the blowing snow algorithm compares the intensity of the signal at the first bin above the surface with a threshold equivalent to 10 times the local molecular backscatter. If the intensity surpasses the threshold, the algorithm identifies the bin as blowing snow with a high concentration of particles in the lowermost 30 m above the surface, i.e. snow particles dislodged and in saltation in the lowest atmospheric levels. Third, the algorithm then interrogates the bins higher in the atmosphere, looking for a large decrease in signal magnitude. This would indicate the top of the blowing snow layer. The search for the top is limited to 500 m above the surface. If the top of the blowing snow layer is not found within that height, it is considered diamond dust. Fourth, the algorithm only considers blowing snow events if the GEOS-5 10 m wind speed exceeds 4 m s$^{-1}$. The vertical resolution of the CALIOP measurements is 30 m, which limits the detection of blowing snow layers of 30 m or greater.

Finally, the updated version of the algorithm (Palm et al., 2017) includes additional checks to screen for clouds in order to avoid low cloud layers being detected as blowing snow. Layers are removed if the maximum of the backscatter profile occurs above 300 m, or if the maximum signal exceeds a predefined threshold.





## 2.2 Ceilometers

Ceilometers are robust ground-based remote sensing instruments, consisting of a vertically pointing laser. The two instruments used in this study are manufactured by Vaisala, but have slightly different characteristics. The first ceilometer, the CL-31, is set up on the roof of Princess Elisabeth station and measures up to 7700 m above ground level ( a.g.l.) (Gorodetskaya et al., 2015)

and the CL 51 at Neumayer station reports measurements up to 13500 m a.g.l. (Konig-Langlo and Loose, 2007). The ceilometer measures the attenuated backscatter profile at a frequency of 15 seconds, with a vertical resolution of 10 meters, used to derive the cloud base height, and information on the cloud phase and cloud cover. The ground based retrievals of blowing snow are derived from the blowing snow detection (BSD) algorithm (for more information, see Gossart et al. (2017)), developed using the ceilometer measurements at Princess Elisabeth station and applicable at other stations (as demonstrated for Neumayer).

The BSD algorithm enables the retrieval of the blowing snow signal from the attenuated backscatter that is measured by the ceilometer.

The BSD algorithm is similar to the blowing snow algorithm for the satellite data, with some modifications. There is no need for a surface detection algorithm and we do not apply any wind threshold. The BSD algorithm consist of two steps. First, it involves the comparison of the near-surface bin attenuated backscatter with the clear-sky threshold. If the backscatter

surpasses the threshold, it indicates the presence of snow particles in the lower atmosphere just above the surface. Second, the BSD requires that the intensity of the signal must decrease with height, which indicates that the concentration of blowing snow particles declines within the layer, until the top of the blowing layer is reached.

Since ceilometers are pointing upward, they discriminate between blowing snow and non-blowing snow, during clear-sky and even in cloudy conditions. In order to detect the presence of clouds (with and without precipitation), the PT algorithm

(Van Tricht et al., 2014) was used, which includes a signal-to-noise ratio correction. The PT algorithm allows the detection of geometrically thicker clouds (> 90 m vertical extent), but does not consider thin clouds (lower limit of optical depth = 0.01). The BSD method has been compared to visual observations at Neumayer III station, and the results agree for heavy blowing snow, but underdetections occur for lighter blowing snow events (Gossart et al., 2017), implying that the ceilometer fails to detect light blowing snow events, or blowing snow that only occurs very close to the ground. This limitation is explained by a

combination of two instrument-specific issues. Firstly, the actual lowest measurement bin of the ceilometer, i.e. the first 15 m above the surface, must be discarded from the analysis because it is polluted by the internal noise of the instrument. Secondly, the ceilometers are installed on the roof of the station, approximately 10 to 15 m above the actual surface. This implies that the ceilometer can only detect blowing snow starting at 30 m above the surface, effectively missing all shallower blowing snow layers.

Since the ceilometer is a ground-based instrument, it is closer to the blowing snow event and is not impeded by the presence of clouds or precipitation. This is an advantage compared to the satellite product, and the reason we consider the ceilometer retrieval as ground truth.





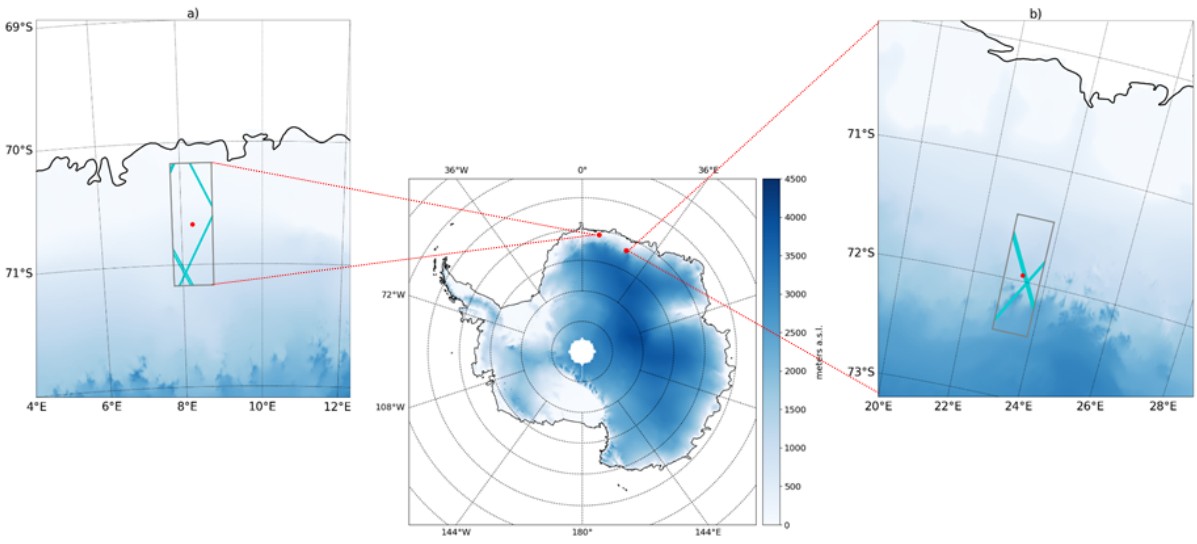

**Figure 1.** Location of the two Antarctic stations used in this study. Center: The Antarctic continent, the red dots indicate the location of the stations. Elevation is represented in blue. Zoom over the Neumayer (a), left) and Princess Elisabeth (b), right) stations. The grey boxes represent the 1 by1 ° area. Elevation is represented in blue and the satellite track is represented by the light blue lines.

| | Princess Elisabeth station | Neumayer III station |
|---|---|---|
| Elevation | 1392 m a.s.l. | 43 m a.s.l. |
| Annual mean air temperature | -18 ° C | - 16 ° C |
| Annual mean wind speed | 5 m s$^{-1}$ | 9 m s$^{-1}$ |
| Annual mean surface wind direction | | |
| – Synoptic disturbances | 90 ° N | 100 ° N |
| – Katabatic conditions | 180 ° N | 170 ° N |
| Annual mean relative humidity | 58% | 90% |
| Annual mean atmospheric surface pressure | 827 hPa | 986 hPa |

**Table 1.** Meteorological conditions at Princess Elisabeth and Neumayer III stations. For extended climatology, see Gorodetskaya et al. (2013) for PE station and Konig-Langlo and Loose (2007) for Neumayer station.



## 2.3 Stations location and data sampling

To compare to satellite retrievals of blowing snow, we use the ceilometer derived blowing snow observations from two ground-based remote sensing ceilometers set up at the East Antarctic stations Neumayer III and Princess Elisabeth. The latter was installed in 2009 as part of the cloud and precipitation observatory (Gorodetskaya et al., 2015; Souverijns et al., 2017, 2018;

Gossart et al., 2017). The Princess Elisabeth station, located at 71.95 ° S and 23.35 ° E, 173 km from the coastline, is shielded by the Sør Røndane mountains (Figure 1). Blowing snow is observed 13% of the summer time by the ceilometer, and mainly associated with warm synoptic regimes (Gossart et al., 2017), related to the transient low-pressure systems passing nearby (Gorodetskaya et al., 2013, 2014).

    The second ceilometer was installed in 2011 at Neumayer III station (Neumayer hereafter), located on the Ekström ice shelf

close to the coast (70.67° S and 8.27° E, Figure 1). Generally, the climate at Neumayer is warmer and windier than at Princess Elisabeth (Table 1). The synoptic origin for blowing snow is also predominant at Neumayer station (Konig-Langlo and Loose, 2007). There, yearly blowing snow frequencies attain nearly 40%, but with substantial seasonal and inter-annual variability (Lenaerts et al., 2009).

    The measurement period of the CL-31 at Princess Elisabeth station starts in February 2010 and extends up to May 2016, but

data is mainly limited to the austral summer season (November to May), as a result of power outages occurring at the station during the Antarctic winter. Only one year (2015) is fully recorded. At Neumayer station, observations are available from 2011 to 2016, without significant data gaps.

    We define 1 by 1°boxes centered around the two stations as the evaluation domain (grey boxes in Figure 1). We consider all satellite overpasses (blue tracks) within the box, for the time period that is concurrent to ceilometer measuring at the station.

Each of these concurrent overpasses is then considered a 'case'. There are 438 cases (or overpasses with coincident data) at Neumayer and 132 at Princess Elisabeth station. Although the CALIPSO satellite repeats the exact same track every 16 days, overpass frequencies and timing differ for both stations (see Table 2). At Princess Elisabeth station, 3 distinct tracks can be considered: the first track passes within a radius of $\approx 3 km$ from the station at 13:55 UTC, while the second track is located $\approx 12 km$ away from the station at 22:25 UTC, and the most distant overpass is located $\approx 66$ km away from the station, at 22:31

UTC. The combination of all tracks results in an overpass frequency every 3rd or 9th day. At Neumayer station, there is no overpass in the vicinity of the station, and there are 4 overpasses every 16 days: the first one at 14:45 UTC on day one at $\approx 35$ km from the station, the second at 23:33 UTC on day 2 within $\approx 23$ km of the station, the third on 14:51 UTC on day 10 within $\approx 44$ km and the 4th at 23:39 UTC on day 11 within $\approx 57$ km, respectively.

    For each case, we consider the full length of the satellite track within the domain. The satellite clear-sky fraction is calculated

by counting the number of surface detections over the total length of the track and dividing by the total number of observations along the track. The cloud coverage fraction is subsequently derived as one minus the clear-sky fraction. Then, the number of clear-sky conditions with and without blowing snow is calculated by determining the number of (no-)blowing snow detections along the clear-sky part of the track. For this purpose, blowing snow occurrence is assumed if the blowing snow confidence level is higher than 1 as described in Palm et al. (2017).



| Day | Hour | Distance (mean) | Number of observations (mean) | Number of cases |
|---|---|---|---|---|
| Princess Elisabeth station | | | | |
| Day 1 | 13:55 UTC | 3.3 km | 216 | 54 |
| Day 4 | 22:31 UTC | 66.0 km | 70 | 24 |
| Day 13 | 22:25 UTC | 11.9 km | 213 | 54 |
| Neumayer station | | | | |
| Day 1 | 14:51 UTC | 43.8 km | 91 | 117 |
| Day 2 | 23:39 UTC | 57.5 km | 24 | 91 |
| Day 10 | 14:45 UTC | 34.3 km | 146 | 118 |
| Day 11 | 23:33 UTC | 23.0 km | 215 | 112 |

**Table 2.** Overview of the overpasses at Princess Elisabeth and Neumayer stations.

To match the spatial sampling of the satellite detections, we define a 20 minute time window during which we consider ceilometer data. The ceilometer measurements are then sub-sampled to the exact moment of satellite overpass plus/minus 10 minutes. Subsequently ceilometer-derived cloudy and clear-sky fractions are calculated from the number of temporal samples indicating the presence/absence of clouds over the time window. In addition, the BSD algorithm is used to determine the

fraction of blowing and no-blowing snow occurring during the clear-sky periods. Since the ceilometer is able to detect blowing snow in cloudy condition, we can also discriminate between cloudy periods with blowing snow, and with no blowing snow.

We then compare, for each individual case, the satellite-based and ground-based blowing snow occurrences, particularly focusing on the fraction of clear-sky and cloudy conditions, as well as the fraction of blowing snow. Statistics of these matches and mismatches are expressed over the whole sample size (total number of overpasses).

For both detection methods, the cloud fraction indicates whether we are in presence of cloudy or clear sky conditions. Since the cloud distribution over the study area is bi-modal and either equal or close to 0 or 100%, we put a 50% threshold for cloud/clear sky detection (see Figure 2).

Regarding blowing snow, Figure 3 indicates that the optimum to minimize errors and maximize agreements, but keeping a minimal number blowing snow events, is reached between 0 and 10 %. We therefore set a threshold for blowing snow at 10%:

blowing snow is assumed if 10% or more of the satellite measurement along track show the presence of blowing snow. For the ceilometer, 10% of temporal detections (or 2 minutes) showing the signature of blowing snow are required.

Then, we define agreements as cases where both the ceilometer and the satellite detect the same cloud and blowing snow occurrences, commission errors when the satellite derived measurement indicates blowing snow, but the ceilometer not, and omission errors if blowing snow is detected by the BSD algorithm, but the satellite product does not indicate the presence of

blowing snow. During the occurrence of optically thick clouds, the satellite is unable to measure beyond the cloud cover and no information can be retrieved about blowing snow occurrences.





**Figure 2.** Cloud fraction at Princess Elisabeth (top) and Neumayer (bottom) stations, the right graph is a zoom between and 10 % for both detection methods.





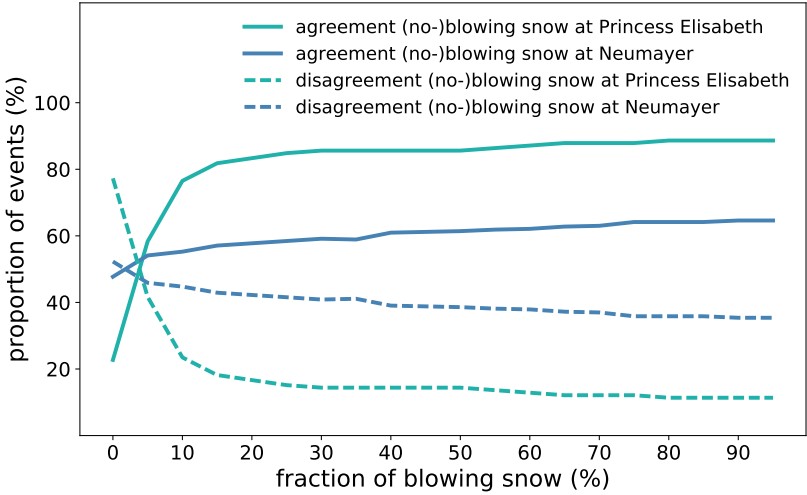

**Figure 3.** Distribution on agreement on (no-) blowing snow (solid lines) and disagreement on (no-)blowing snow (dashed lines) at Princess Elisabeth station (green) and Neumayer station (blue).

## 3  Results

The percentages in the following section refer to the whole sample size (132 'cases' at Princess Elisabeth, and 438 'cases' at Neumayer station), also for commission and omission errors, if not indicated otherwise.

### 3.1  Fraction of blowing snow and clouds

5  Figure 4 illustrates the blowing snow fraction as detected by both the BSD and the satellite algorithms at Princess Elisabeth and Neumayer stations. The in-situ observations show that blowing snow occurs 13% and 38% of the time at Princess Elisabeth (mainly limited to summer-only) and Neumayer station (annual), respectively. However, the satellite detects only two of these blowing snow events at Princess Elisabeth, and all blowing snow cases are missed at Neumayer. These omission errors (blowing snow detected by the ceilometer algorithm, but not the satellite, over all measurements) are more frequent at Neumayer station

10  (38% of all overpasses, and 100 % of blowing snow cases) than at Princess Elisabeth station (12% of all overpasses, and 90% of all blowing snow cases). This implies that the satellite slightly overestimates summer blowing snow frequency at Princess Elisabeth station, but underestimates yearly blowing snow frequency by almost one third at Neumayer station, essentially due to the cloud cover limiting the satellite detection. In contrast, commission errors (blowing snow detected by the satellite algorithm, but not the ceilometer) reach 10% at Princess Elisabeth station, and are negligible at Neumayer station (4%).

15      At Princess Elisabeth station, clear-sky conditions occur more frequently than cloudy conditions (Figure 2). The agreement on clear-sky conditions (35%) or cloudy (51%) dominates, while the satellite omission of clouds reaches 9% of the cloudy and identifies 5% of clear sky conditions as clouds (commission error), respectively. At Neumayer, cloudy conditions dominate





**Figure 4.** Blowing snow fraction at Princess Elisabeth (top) and Neumayer (bottom) stations, the right graph is a zoom between 0 and 10 %
for both detection methods.





(73%). The agreement between satellite and ceilometer clear sky reaches 20% and 34% for cloudy conditions. Figure 2 shows that some cloudy events are identified as cloud-free by the satellite (39%). In addition, 7% of clear- sky conditions is classified by the satellite as cloudy conditions.

## 3.2 Satellite commission and omission errors

In this section, we give an overview of the over-estimations and under-estimations of blowing snow frequency, as well as cloudy conditions, by the satellite (Figure 5). First, a significant part of the blowing snow occurring during cloudy conditions is missed by the satellite algorithm. These omissions occur because the satellite attenuated backscatter signal does not penetrate through the cloud and the satellite only detects a cloud (Fig.5, image B). This satellite under-detection of blowing snow represents 13 and 21% of all cases at Princess Elisabeth and Neumayer stations, respectively, which implies that if there is a cloud, in 30%

of the cases, they are accompanied by blowing snow at Princess Elisabeth and Neumayer stations.

At Neumayer only, in 18 % of the cases, the satellite algorithm indicates clear sky, while a cloud is detected in the ceilometer attenuated backscatter signal (Fig. 5, image A). Since thin clouds are identified by the ceilometer, at optical depths of minimum 0.01, and the satellite signal is blocked by thick clouds only (optical depth of 2-3), mismatches can occur when the satellite sees through a cloud, as identified by the ceilometer, if it is a thin cloud. Alternatively, at Princess Elisabeth station 8% of

the cases are signalled as blowing snow by the satellite retrieval algorithm while the ceilometer indicates clear sky but no blowing snow (Fig.5, image C). Overall, omission errors for blowing snow reach 14% at Princess Elisabeth station, and 39% at Neumayer station, when the signal encounters a cloud impeding the detection, or in the aforementioned cases. Disagreement between ceilometers indicating the presence of clouds while the satellite indicates a ground return count for 7% at Princess Elisabeth, and 18% at Neumayer station. The opposite, clear sky as detected by the ceilometer and cloud for CALIPSO occur

in 6% and 10% at Princess Elisabeth and Neumayer, respectively. As discussed in section 4, plausible reasons include 1) spatial inhomogeneity of blowing snow, satellite 2) signal sensitivity resulting in missing blowing snow, 3) misinterpreting low cloud or fog as blowing snow and 4) the bin identified as the first bin above the ground by the satellite contains some surface signal which is then interpreted as blowing snow. The latter would happen only over very rough terrain and mountains.

Since the detection of blowing snow is impeded by the presence of thick clouds, the frequencies derived from the satellite

retrievals are realistic for clear-sky or thin cloud conditions only. The ceilometer retrieval illustrated in Figure 6 shows the probability density function of blowing snow fraction for the ceilometer for both cloudy and clear-sky conditions. At Princess Elisabeth station, if we consider 10% as the blowing snow threshold (2 minutes), the probabilities are lower for blowing snow under cloud cover than under clear sky conditions. This is also the case at Neumayer station. However, the magnitude (size and duration of the event) is in general larger for blowing snow accompanied with clouds than for clear sky conditions: the

clear-sky blowing snow events are shorter in time (less than 10 minutes): around 20% and up to 40% of the measurement time period at both stations. In cloudy conditions, however, longer blowing snow events are more frequent and peak around a bit more than 10 minutes (or 50-60% of the time window) and numerous events reach 20 minutes (90-100%). A similar situation is also observed at Neumayer station.





**BLOWING SNOW OMISSION**          **BLOWING SNOW COMISSION**

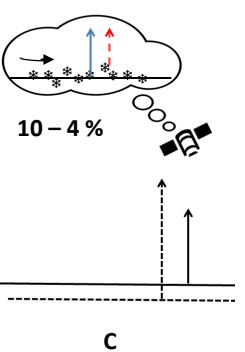

1 − 18 %          13 − 21 %          10 − 4 %

A          B          C

**CLOUD/SURFACE MISINTERPRETATION**

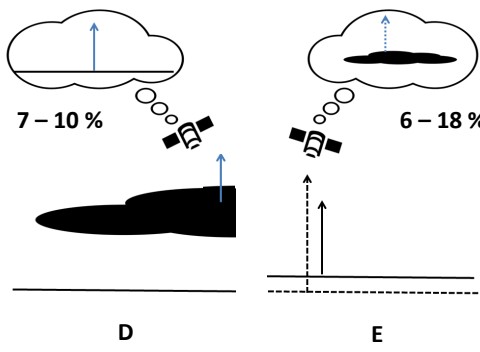

7 − 10 %          6 − 18 %

D          E

**Figure 5.** Possible type of blowing snow commission/omission and cloud/clear sky mis-classification. Upper left image: the ceilometer-based BSD detects a blowing snow layer overlaid by clouds, while the satellite detects clear sky (A) or sees only the cloud (B) and effectively misses the blowing snow. Upper right: the satellite algorithm detects a blowing snow layer unseen by the ceilometer (possibly due to the pollution of the first bin above ground, misinterpretation of the ground return or of a low cloud or fog as blowing snow, C). Lower images: the satellite and ceilometer detect the same absence/presence of blowing snow but the satellite indicates clear sky while the ceilometer detects a cloud (D), or detects a cloud while the ceilomteter indicates clear sky (E). The percentages refer to Princess Elisabeth - Neumayer, respectively.



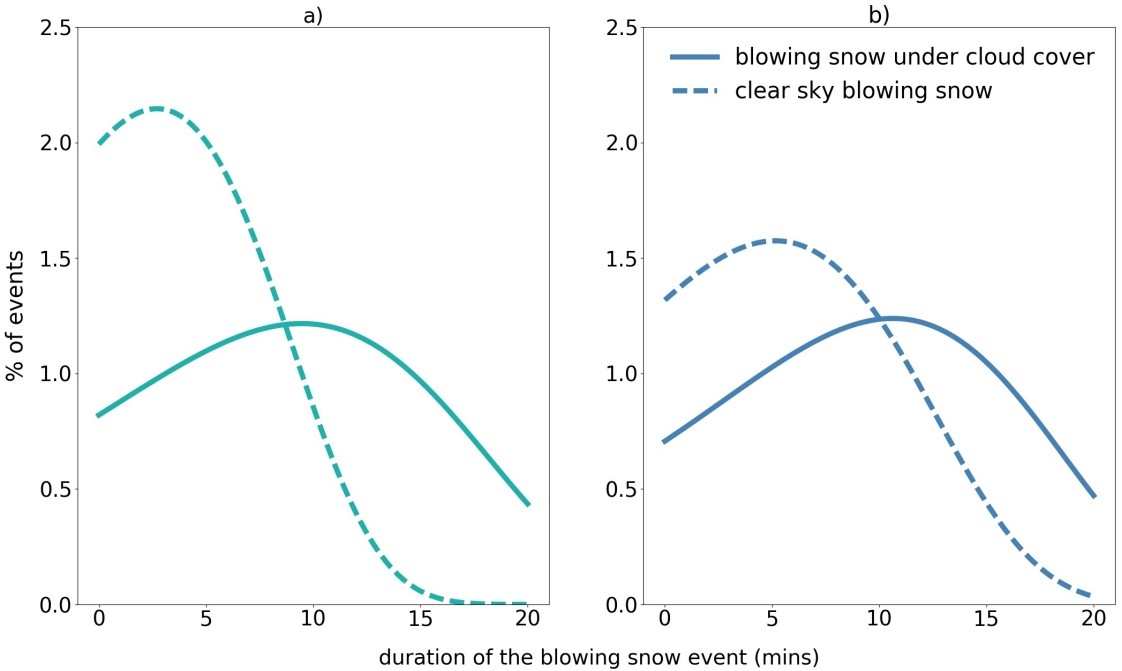

**Figure 6.** Probability density function for blowing snow under cloud cover (solid) or under clear sky (dashed) conditions, using the ceilometer-based BSD algorithm at Princess Elisabeth and Neumayer stations.

## 4 Discussion

Our analysis has shown that, while the satellite-derived product is a useful tool to estimate blowing snow frequencies over the entire AIS, it is limited to clear sky or thin cloud conditions which are predominate over the interior of East Antarctica. In coastal areas and most of West Antarctica, opaque clouds are often present.

5     Clear sky blowing snow is rare at both stations, resulting in blowing snow detection by the ceilometer essentially occurring under cloudy conditions (95% of all blowing snow detection). Because the satellite is limited to clear sky, only 2 events are detected by both methods at Princess Elisabeth station, and none at Neumayer. However, taking the agreement on the absence of blowing snow into account leads to agreement of the two methods of 78% at Princess Elisabeth, and limits the omission and commission errors to ≈ 10%. At Neumayer, the total agreement on presence and absence of blowing snow reaches 57%.

10  Omission errors reach 38%, but are all due to the presence of clouds limiting the measurements of blowing snow. Palm et al. (2017) estimates the error associated with the presence of thick clouds as a 50% occurrence of blowing snow, leading to 25-30% increase on blowing snow occurrence over coastal areas of East Antarctica. However, our results show that at most 38% of the cloudy cases are accompanied with blowing snow at Neumayer station, leading to potential overestimation of the error due to the presence of clouds in blowing snow detection from the satellite.



Mismatsches are observed between cloud and clear sky classification. The differences in the two sensors could explain some of the mismatch: e.g. thin clouds measured as clouds by the ceilometer (lower limit of optical depth= 0.01) but are not thick enough to totally attenuate the satellite backscatter signal (this only happens when the cloud has an optical depth greater than about 2-3). The ceilometer may be detecting clouds while CALIPSO does not detect this cloud. This can increase the total

number of cloudy conditions and lower the frequency of blowing snow occurrences under cloud cover, compared to the satellite estimate. Since storms are frequently associated with blowing snow, but not thin or high clouds, a means to distinguish between the cloud types could help constrain the satellite-derived probability of blowing snow under cloud cover, by considering mainly the stormy conditions.

While the satellite omission errors associated with the presence of clouds most likely lead to under-estimations of the

blowing snow frequency at both stations, the blowing snow commission errors during clear sky is the predominant reason for blowing snow over-estimation at Princess Elisabeth station. This is most likely due to spatial inhomogeneity of blowing snow, especially in the case of a CALIPSO track at the edge of the domain. In addition, it is possible that the satellite falsely detects the presence of blowing snow by mistaking a portion of the surface signal for blowing snow. Since Princess Elisabeth is located at the foot of the Sør Røndane mountains (Figure 1), and these commission errors are not observed at Neumayer station, such

cases could arise. Overall, taking into account the omission and these omission errors, the satellite blowing snow detection rate reaches 11%, similar to the 12% detected by the ceilometer at Princess Elisabeth station. At Neumayer, however, there is no compensation between commission and omission errors, which leads to a substantially lower blowing snow rate detected by the satellite (4%) than indicated by the ceilometer (around 40%), predominantly due to the presence of cloud decks. Table 3 shows that blowing snow occur predominantly in cloudy conditions at both Princess Elisabeth and Neumayer stations. Overall,

the satellite blowing snow frequencies are roughly in line with the ceilometer clear sky values.

In addition, comparing a point measurements on the ground to a satellite track within a 1 by 1° grid is challenging and implies the set up of thresholds. This might explain part of the differences between the two detection results (see table 3), where blowing snow frequencies are measured by the two methods independently. Blowing snow is detected by the ceilometer if exceeding 20 minutes, and the satellite detects blowing snow as each ground detection with a blowing snow confidence level

over 1. Moreover the ceilometer at Princess Elisabeth station is limited mainly to summer data, where the satellite retrievals during the summer are very sparse, especially in this area.

Finally, the low frequency of winter blowing snow observed at Princess Elisabeth is due to its particular location. The station happens to be right in a large gradient of blowing snow frequency, leading to low blowing snow frequencies, even in winter. While clear sky winter blowing snow frequency detected by the satellite is limited to 7.5% at Princess Elisabeth, the next 1 by

1° box South of the station shows satellite detected blowing snow frequencies of 50% for the winter 2015. Such local variations in blowing snow frequency patterns highlight the utility of point measurements and the difficulty to compare spatial averages to local point measurements.

Finally, we defined a 10% threshold for blowing snow on both the satellite and ceilometer results, which leads to a rather good agreement between the two methods. However, published blowing snow frequencies, transport and sublimation rates

(Palm et al., 2011, 2017) display results where tracks are sub-sampled on a 1 by 1° grid, and each ground detection is an





| Observation | Season | % cloudy blowing snow | % clear sky blowing snow |
|---|---|---|---|
| ceilometer at Princess Elisabeth | Dec - Feb | 10.1 | 0.4 |
| | Jun - Aug | 15.8 | 2.3 |
| satellite at Princess Elisabeth | Dec - Feb | X | 5.6 |
| | Jun - Aug | X | 7.5 |
| ceilometer at Neumayer | Dec - Feb | 30.6 | 4.3 |
| | Jun - Aug | 43.7 | 3.5 |
| satellite at Neumayer | Dec - Feb | X | 2.2 |
| | Jun - Aug | X | 2.8 |

**Table 3.** Presentation of the climatology of blowing snow at Princess Elisabeth and Neumayer stations, as retrieved from the two different sensors individually, for the 2011-2016 time period (the whole time series is considered for each sensor, no time collocation is applied). Note that the ceilometer at Princess Elisabeth station winter percentage is for one year only (2015).

'observation'. Blowing snow detection along the satellite track corresponds then to a CALIOP observation complying with the blowing snow detection algorithm defined in section 2.1. Subsequently all blowing snow detections are summed and divided by the number of ground detections within the 1 by 1° cell. In this case, the detection threshold is lowered to only one measurement along track (mean track length varies from 24 to 216 observations). If this method gives acceptable results for continent-wide or regional estimations, it most probably leads to commission errors if validated with ground truth at areas near the coast on the AIS.

## 5 Conclusions

In this paper, we compare the space borne blowing snow retrieval based on the CALIOP lidar on board the CALIPSO satellite with blowing snow retrievals from attenuated backscatter profiles from ground-based ceilometers set up at two stations in East Antarctica. We define a 1 by 1° grid over both stations and investigate each satellite overpass within the predefined domain to concurrent ceilometer data within a 20 minute time window. The measurement period extends from 2011 to 2016 year-round at Neumayer and is limited to summer only at Princess Elisabeth station. We compare the correspondence in clear sky blowing snow detection and cloud detection by the satellite with the Blowing Snow Detection (BSD) algorithm, based on the ceilometer attenuated backscatter profile.

First, cloud decks shield blowing snow events from the satellite, whereas the ceilometer can identify blowing snow during cloudy conditions. This is not trivial, since 95% of the blowing snow occurs under cloud cover at Princess Elisabeth and Neumayer stations. Then, blowing snow commission errors equal 10% at Princess Elisabeth station, probably due to the inhomogeneity of the event and/or the bin identified as the first bin above the ground contains some surface signal which is then interpreted as blowing snow, which can happen in rough terrain. Such commission errors are absent at Neumayer.





However, while the agreement between the satellite and ceilometer-derived cloud fraction is good at Princess Elisabeth station it is not the case at Neumayer station. There, many of the clouds detected by the ceilometer are likely transmissive. In such cases the satellite would detect the ground return and they would be classified as clear sky.

Finally, two issues are identified. Firstly, the role of precipitation in blowing snow occurrence : cloudy conditions are fre-
quently associated with snowfall events at the stations, and the freshly fallen snow contributes to blowing snow. Secondly, the comparison developed in this paper is limited to two stations where blowing snow is significantly driven by synoptic events (occurs only/mostly during cloudy conditions), which is not the case for the entire AIS. We therefore advise to compare the satellite algorithm to more inland stations, where the influence of clouds is reduced.

Our analysis shows that the satellite blowing snow retrieval is a useful product but further investigation is needed to reduce
the uncertainty on blowing snow frequencies under cloud cover. In such areas, it may be important to differentiate cloud type or height to help estimate the likelihood of blowing snow occurrence. Investigating the difference in cloud classification for both sensors could further constrain mismatches between the two detection methods. Plus, while putting a 10% threshold gives good agreement between the two detection methods on blowing snow frequencies, it can lead to over-estimations when trying to compare point locations to blowing snow frequencies, sublimation and transport rates from previous studies using a 1 by 1°
grid.

*Code availability.* the Blowing Snow Detection algorithm is freely available by contacting alexandra.gossart@kuleuven.be.

*Data availability.* Neumayer station datasets are publicly available on the Pangaea portal (König-Langlo, 2011, 2012, 2013, 2014, 2015, 2016, 2017) and data from the instruments at Princess Elisabeth station are available on the www.aerocloud.be website. CALIPSO blowing snow data are now available at the NASA Langley Atmospheric Science Data Center (ASDC) at https://earthdata.nasa.gov/about/daacs/daac-
asdc.

*Competing interests.* The authors declare no competing interests.

*Acknowledgements.* We thank the logistic teams and the Royal Meteorological Institute for executing the yearly maintenance of our instruments at the Princess Elisabeth station and for their help by installing and setting up the new AWS. We thank Wim Boot, Carleen Reijmer, and Michiel van den Broeke (Utrecht University, Institute for Marine and Atmospheric Research Utrecht) for the development of the automatic
weather station, technical support and raw data processing. This work was supported by the Belgian Science Policy Office (BELSPO; grant number BR/143/A2/AEROCLOUD) and the Research Foundation Flanders (FWO; grant number G0C2215N) The CALIPSO blowing snow algorithm and data analysis was performed under NASA contracts NNH14CK40C and NNH14CK39C. The authors would like to thank NASA program managers Thomas Wagner and David Considine for their support and encouragement.



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
