# Peer review of "Blowing snow in East Antarctica: comparison of ground-based and space-borne retrievals"

_The Cryosphere, 2019_

## Referee Comment (RC1) · Anonymous Referee #1 · 18 Apr 2019

In the manuscript, the authors compare the results of blowing snow detection of their recent papers using satellite retrievals from CALIPSO (Palm et al., 2011 and 2017) with novel ground-based ceilometer observations (Gossart et al., 2017), in order to estimate/validate/compare satellite-based retrievals with a point measurement on the ground ceilometer of blowing snow.

Although the purpose of manuscripts is of great interest for the surface mass balance study, the manuscript is well written and the methlogies are robust, the analysis in the manuscript suffers from major issues which make its results insufficient to be published on The Cryosphere journal.

The first issue is that the authors use two Stations where the katabatic blowing snow phenomena are limited by their geographic location and associated climatic condition: Newmayer is located on the ice shelf, far from katabatic wind flow where blowing snow phenomena is mainly linked to synoptic storm, whereas Princess Elisabeth station is located at strong change of topographic slope surrounded wide ranging blue ice area with extensive snow sublimation process during katabatic blowing snow event. Moreover the data available for Princess Elisabeth station are limited to the summer season when the katabatic and blowing snow event are very limited or nul.

These topographic/climatic features limited the number of events detected by both methods, only 2 at Princess Elisabeth station, and none at Neumayer, as reported by Authors. The amount of the data to compare are not significant and limited strongly the value of results.

---

## Referee Comment (RC2) · Anonymous Referee #2 · 10 May 2019

In this article, Gossart and co-authors evaluate a satellite product of blowing snow detection based on the lidar CALIOP on board CALIPSO since 2006, which is compared with blowing snow events derived from ground-based ceilometers at Princess Elizabeth and Neumayer stations for the 2011-2016 period. Grid boxes of 1°x1° around the stations are defined where each satellite overpass is considered as a "case", and the authors set up a threshold of 10% or more of the satellite measurement along track for defining the presence of blowing snow.

The stations are located at a distance of around 1000 km from each other. Neumayer is closer to the sea (43 m a.s.l.) whereas Princess Elizabeth is located at the ice sheet

flank (1392 m a.s.l.) and shielded by nunatak mountains.

The two blowing snow detection products are already published : Palm et al. (2017, TC) for blowing snow transport derived from CALISPO, and Gossart et al. (2017, TC) for blowing snow detection with ceilometers at Princess Elizabeth and Neumayer.

An already known issue with satellite detection of blowing snow is that it does not detect blowing snow events under cloudy conditions (Palm et al., 2017). And it is also already known that most of blowing snow events at the two stations occur under cloudy conditions (92 % in Gossart et al., 2017). Here 95% of the events of blowing snow detected by ceilometers occur under cloudy conditions. This given, and despite the undoubtable work of the authors, Princess Elizabeth and Neumayer stations don't seem right locations to evaluate the satellite product. I don't see how the paper could be improved as the only point that can be assessed is whether the satellite product is able to *not* predict blowing snow under clear sky conditions, which the authors did but which I don't think is sufficiently of interest to be published in The Cryosphere. Maybe the paper can be redirected in improving cloud detection in CALIPSO?

In this view, I acknowledge the work of the authors, but I think because of the major issue in the ground of the paper described above, this article is not suitable for publication in The Cryosphere.

---

## Editor Comment (EC1) · Tobias Sauter (Editor) · 28 May 2019

Dear colleagues,

the submitted paper was reviewed by two independent reviewers and I would like to take this opportunity to thank both reviewers for their thoughtful reviews.

Both reviewers expressed fundamental concerns and concluded that the submitted article did not meet the requirements for publication in 'The Cryosphere'. The doubts of the expert opinions concern on the one hand the choice of the locations and on the other hand the insufficient satellite detection of snow drift under cloudy conditions. I

support the comments and doubts of the two experts and do not see how they can be dispelled in the present version. Based on the opinions and my own assessment, I am sorry to inform you that I am not asking you to resubmit a revised version.

As already noted by Reviewer 1, the work is of great relevance for surface mass balance studies and the comments can help increase awareness and impact in the community. In this context, I hope that the reviews and comments will make a good contribution to an improved version of the study, for which you are welcome to consider TC as a publication platform again.

Tobias Sauter

---

## Author Comment (AC1) · 3 Jun 2019

We thank the two Referees and the Editor for their comments on our paper. We regret that the manuscript was not considered for publication to The Cryosphere Journal.

We believe that this paper presents the first comparison of blowing snow retrievals by the CALIPSO satellite with ground-based observations from ceilometers. The satellite algorithm, blowing snow frequencies, transport and sublimation fluxes, as well as blowing snow climatology have been published in three different papers (Palm et al., 2011, 2017 and 2018) and a book (Palm et al.,2019). These products are extremely valuable and widely used by the cryospheric community since they offer a continental wide assessment of blowing snow, which is an important but unknown component of the surface mass balance of the Antarctic ice sheet. Therefore a comparison with ground truth is – in our view – of very high importance. The second reviewer acknowledges the importance of the study. The main argumentation of both reviewers is that Princess Elizabeth and Neumayer station are not the right location for doing this evaluation, due to the frequent occurrence of blowing snow in cloudy conditions. Indeed, we already acknowledged in the original manuscript the frequent occurrence of blowing snow under cloudy conditions at these sites.

However, we do not think this would be a reason to reject the paper due to the reason outlined below:

Blowing snow during precipitation and cloudy conditions is a frequently occurring phenomenon (not only at these two sites) and therefore an important contributor to the snow transport and sublimation over the entire Antarctica: While the reviewers say this has been already in Gossart et al 2017 paper, the present paper extends the work comparing to the satellite-based algorithm, thus highlighting the potentially important process of blowing snow during cloudy/precipitation for the satellite-derived products. Of course the limitation of satellite detection to clear sky conditions is an already known issue, but a quantification of this effect has not been performed earlier. Therefore, our intent here is to quantify the percentage of time that blowing snow occurs during cloudy conditions that result in the satellite not being able to render a decision. Moreover, the paper demonstrates that blowing snow frequencies under clear sky conditions are not at all representative for the all sky blowing snow frequencies. These results can subsequently be used in further studies to improve the satellite estimates of blowing snow under clouds. This has not been addressed in the Gossart et al., 2017 paper. A second point raised by reviewer 1 is that the number of events detected at both station is too low (2 at Princess Elisabeth and none at Neumayer). However, this is only the number of events detected by both methods (satellite and ceilometer). There are in total 438 and 130 comparisons at Neumayer and Princess Elisabeth stations, respectively, which

[Figure]

did not result in concurrent detection of blowing snow. Some were blocked by clouds, some were disagreements. We therefore do not agree with this point raised by the reviewer. Based on these 438 and 130 comparisons, the important findings - unrelated to the cloud issue – were presented in the paper: A percentage of 10% (4%) of the measurements at Princess Elisabeth (and Neumayer) are identified as blowing snow by the satellite but not by the ceilometer – relevant information for the remote sensing and the user community in our opinion. This indicates that the satellite retrieval can be hampered in the case of rough terrain (at Princess Elisabeth station). The paper shows that cloud detection remains a research challenge for deriving blowing snow conditions from space: At Princess Elisabeth (Neumayer), during 6% (18%) of the cases, the satellite detects a cloud which is not detected by the ceilometer and during 8% (28%) of the cases the ceilometer detects a cloud which is not detected by the satellite. In retrospect, perhaps indeed we framed this work too much as an Antarctic wide validation of the satellite algorithm and re-framing would be needed, but we do strongly feel that it would be very regrettable not to use the ceilometer ground truth for improving the satellite products in this very sparse data region. At the moment the ceilometer-derived blowing snow products are only available for Neumayer and Princess Elisabeth station. In the future, - as already argued in the paper – the methodology of this paper could also be used in interior regions once more ceilometer-derived blowing snow products would become available to identify how well the satellite products perform for regions that have less influence of synoptics systems.

References

- Palm, S. P., Yang, Y., Spinhirne, J. D., and Marshak, A. (2011) Satellite remote sensing of blowing snow properties over Antarctica, Journal of Geophysical Research: Atmospheres, 116

- Palm, S. P., Kayetha, V., Yang, Y., and Pauly, R. (2017) Blowing snow sublimation and transport over Antarctica from 11 years of CALIPSO observations, The Cryosphere, 11, 2555– 2569.

- Palm, S. P., Kayetha, V. and Yang, Y. (2018) Toward a Satellite-Derived Climatology of Blowing Snow Over Antarctica, Journal of Geophysical Research : Atmospheres, 123(18), 10- 301 – 10-313.

- Palm, Stephen and Yang, Yuekui and Kayetha, Vinay (2019) New Perspectives on Blowing Snow in Antarctica and Implications for Ice Sheet Mass Balance, in Antarctica, a Key to Global Change, Ed. Masoki Kanao, DOI: 10.5772/intechopen.75265, ISBN: 978-1-78985-816-7
* * *